# Modelling interference between vectors of non-persistently transmitted plant viruses to identify effective control strategies

**Marta Zaffaroni**[1¤], **Loup Rimbaud**[2], **Ludovic Mailleret**[3,4], **Nik J. Cunniffe**[5☯*], **Daniele Bevacqua**[1☯]

**1** INRAE, UR1115 Plantes et Systèmes de culture Horticoles (PSH), Site Agroparc, Avignon, France, **2** INRAE, UR0407 Pathologie Végétale, Domaine St Maurice, Montfavet, France, **3** Université Côte d'Azur, Inria, INRAE, CNRS, Sorbonne Université, Biocore team, Sophia Antipolis, France, **4** Université Côte d'Azur, INRAE, CNRS, ISA, Sophia Antipolis, France, **5** Department of Plant Sciences, University of Cambridge, Cambridge, United Kingdom

☯ These authors contributed equally to this work.
¤ Current address: INRAE, Bordeaux Sciences Agro, ISVV, SAVE, F-33140, Villenave d'Ornon, France
* njc1001@cam.ac.uk

**Data Availability Statement:** All relevant data are within the manuscript and its Supporting information files.

## Abstract

Aphids are the primary vector of plant viruses. Transient aphids, which probe several plants per day, are considered to be the principal vectors of non-persistently transmitted (NPT) viruses. However, resident aphids, which can complete their life cycle on a single host and are affected by agronomic practices, can transmit NPT viruses as well. Moreover, they can interfere both directly and indirectly with transient aphids, eventually shaping plant disease dynamics. By means of an epidemiological model, originally accounting for ecological principles and agronomic practices, we explore the consequences of fertilization and irrigation, pesticide deployment and roguing of infected plants on the spread of viral diseases in crops. Our results indicate that the spread of NPT viruses can be *i*) both reduced or increased by fertilization and irrigation, depending on whether the interference is direct or indirect; *ii*) counter-intuitively increased by pesticide application and *iii*) reduced by roguing infected plants. We show that a better understanding of vectors' interactions would enhance our understanding of disease transmission, supporting the development of disease management strategies.

## Author summary

A range of both experimental and theoretical studies show that the behaviour and population dynamics of insects depend strongly upon interactions with other insect species. These interactions have the potential to greatly affect the dynamics of insect-vectored plant disease, as transmission of viruses is intimately dependent on the local density of vectors, as well as how they select and move between potential host plants. Surprisingly, the effects of interaction between vector species on epidemics remains little studied and even worse understood, probably because experimentation is costly and difficult. Here, we present a model which permits us to investigate the effect of interaction between a

**Funding:** The PhD grant to M.Z. was funded by the PACA (Provence-Alpes-Côtes d'Azur) region and INRAE Agroècosistèmes department. M.Z. also thanks Avignon Université for the Bourse Perdiguier, which supported her temporary stay at the University of Cambridge during which much of this work was done. The funders had no role in study design, data collection and analysis, decision to publish, or preparation of the manuscript.

**Competing interests:** The authors have declared that no competing interests exist.

virus, two vector species and the host plant on the spread of viral disease in crops. In this study, our model is used to explore the consequences of common agronomic practices on epidemics. Our study highlights the importance of exploring vectors' interactions to enhance the understanding of disease transmission, supporting the development of disease management strategies.

## Introduction

Aphids transmit nearly 30% of known plant virus species [1, 2]. Aphids vector the majority of non-persistent transmitted (NPT) viruses with virus particles (virions) remaining loosely attached to the insect's stylets [2, 3]. According to this transmission mode, virions are rapidly acquired from infected plants, briefly retained by their vector and inoculated to healthy plants during plant sampling probes [3, 4]. NPT viruses are responsible for severe damage to crops [5]. For instance, *Plum pox virus* (PPV), which is vectored by more than 20 aphid species worldwide, is responsible for sharka, the most devastating disease of stone fruit trees [6]. *Potato virus Y* (PVY), which is spread by more than 50 aphid species, threatens the production of a range of solanaceous crops, including potato, tomato, tobacco, and pepper [7].

The epidemiology of NPT viruses is closely related to the behaviour of aphid vectors, in particular to *i*) aphids' ability to acquire and inoculate the virus during sampling probes and *ii*) their propensity for moving among plants [8]. With respect to a given plant host species, aphid species can be classified as: "residents", which under favourable conditions, spend most of their life on the same host plant individual, or "transients", which land and probe numerous plant individuals in the same day [9, 10]. Although transient aphids are commonly considered the principal vectors of NPT viruses, resident aphids can also efficiently transmit NPT viruses when they are induced to change their host, for example in response to crowding or to changes in plant nutrient contents [11]. For instance, the green peach aphid *Myzus persicae*, despite being a resident aphid species that colonises peach *Prunus persica* and potato *Solanum tuberosum* plants [9, 12], was observed to be an efficient vector of two NPT viruses (PPV and PVY) at laboratory conditions [6, 12, 13]. In addition, the presence of resident aphids may affect transient aphids' behaviour [14–16]. For example, in their experiment with three aphid species, Mehrparvar and colleagues [17] showed that aphid presence on a plant discourages other aphid species to visit the same plant.

Different mechanisms of interference might characterize interactions between resident and transient aphid species. Resident aphids can interfere *i*) directly through the production of pheromones that can have a repelling effect towards other aphid species [9]; or *ii*) indirectly, by inducing the host plant to produce volatile compounds as a defensive mechanism, which may lower plant attractiveness to other aphid species [18]. Such interference mechanisms are likely to reduce the number of plants visited by transient aphids in a given area and increase their propensity for leaving the area [19, 20].

As far as we know, the effect of the interference between resident and transient aphids on the spread of NPT viruses has never been explored. Understanding NPT viruses spread is complex because experimentation is costly and difficult: symptoms may be difficult to detect and experimental trials in the vicinity of susceptible commercial crops may be restricted [21, 22]. Mathematical models are thus particularly useful to provide complementary insights on virus spread [23], and to design and test management strategies, while circumventing the difficulties associated with experiments (*e.g.* [13, 24, 25]). Numerous models have been developed to study the role of vector population dynamics and vector-host-pathogen interactions on the

spread of NPT viruses (*e.g.* [26–33]). Shaw and colleagues [28] developed a model to assess the contributions of vector life history traits (*e.g.* growth rates, fecundity, and longevity) and behavior (*e.g.* vector preferences for settling and feeding) to pathogen spread. Crowder and colleagues [32] developed a model where the vector life history traits and behaviour were varied to explore the effect of interaction (*e.g.* predation, competition and mutualism) between a vector and a non vector species on the spread of plant pathogens. However, all existing studies consider only a single vector species per virus, which limits the possibility to assess the effects of interference between two or more vector species.

In the present work, we develop a general epidemiological model which describes the temporal variation of the number of susceptible and infected plants, and of the number of non viruliferous and viruliferous resident and transient aphids in a single field. We apply the model to explore the role of inter-specific interference upon resident and transient aphid behaviour and the resulting effects on the invasion, persistence and control of NPT viruses in agroecosystems. We use the model to analyze the effects of common agricultural practices, such as fertilization and irrigation, pesticide application and roguing, upon the spread of NPT viruses. We apply the model to a general pathosystem composed of a NPT virus vectored by a resident and a transient aphid species.

## Materials and methods

### Model outlines and assumptions

The model is schematically represented in Fig 1.

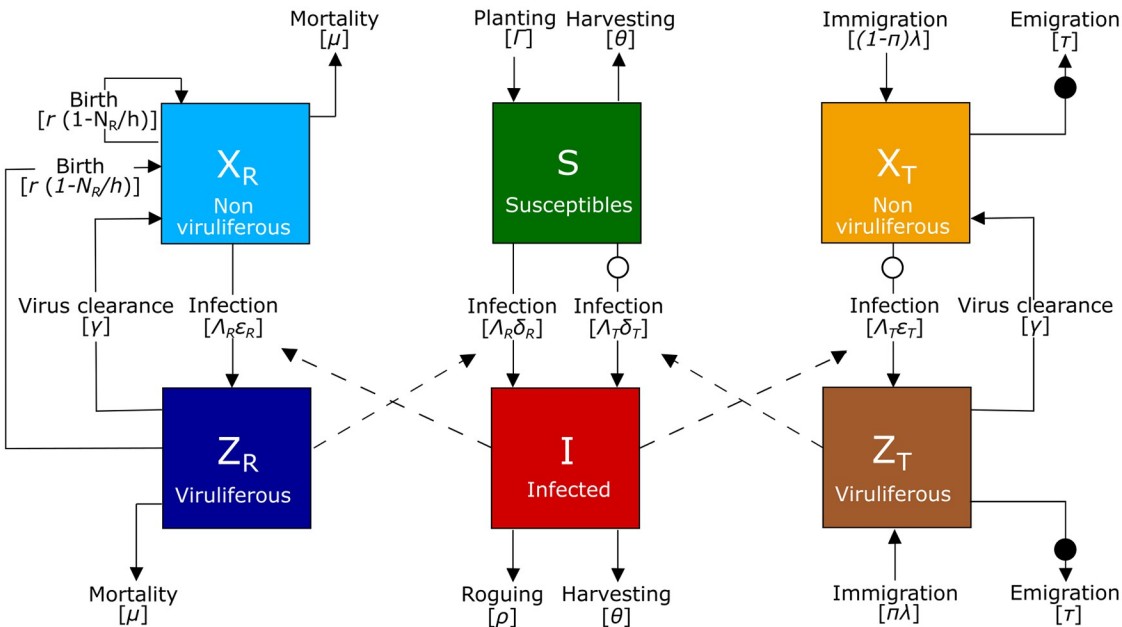

**Fig 1. Single host-multi vector model.** Schematic representation of the single host-multi vector model, where the total number of host plants is partitioned into susceptible ($S$) and infected ($I$) individuals. Aphids are partitioned into non viruliferous ($X_i$) and viruliferous ($Z_i$), and are classified as resident ($i = R$) or transient ($i = T$). Dashed arrows identify the contacts between viruliferous aphids and susceptible plants, and between infected plants and non viruliferous aphids, which affect the infection rates. Circles identify the processes affected by inferences exerted by resident towards transient aphids (visiting interference in white and emigration interference in black). The total number of plants per hectare is $N_P = S + I$, the average number of resident aphids per plant is $N_R = X_R + Z_R$ and the average number of transient aphids visiting a plant per unit time is $N_T = X_T + Z_T$. Details on the processes involved are given in the main text.

The plant population (plant ha$^{-1}$) is structured into two compartments: susceptible ($S$) and infected ($I$). Both resident (subscript R hereafter) and transient (subscript T hereafter) aphid populations (aphid plant$^{-1}$) are structured into non viruliferous ($X_R$ and $X_T$) and viruliferous ($Z_R$ and $Z_T$). The total number of plants per hectare is $N_P$ ($N_P = S + I$), the average number of resident aphids per plant is $N_R$ ($N_R = X_R + Z_R$) and the average number of transient aphids per plant is $N_T$ ($N_T = X_T + Z_T$). A susceptible plant can be infected if it enters in contact with a viruliferous aphid, resident ($Z_R$) or transient ($Z_T$). The probability per unit time that a susceptible plant becomes infected (*i.e.* force of infection) depends on i) the rate of contact between plant and aphid $\Lambda_i(X_i + Z_i)$ ($i = R, T$) where $\Lambda_i$ is the number of plants visited by an aphid per unit time and ($X_i + Z_i$) is the average number of aphids per plant, ii) the probability $\frac{Z_i}{X_i+Z_i}$ that the contact is indeed with a viruliferous aphid and iii) the probability $\delta_i$ that the contact leads to virus inoculation. Therefore the number of infected plants per unit time at the plant population scale is $\sum_{i=R,T} \Lambda_i \delta_i Z_i S$.

We assumed that infected plants are removed at a *per capita* rate $\rho$ and that both susceptible and infected plants are harvested at a *per capita* rate $\theta$. Furthermore, $\Gamma$ new susceptible plants are planted per unit time. A non viruliferous aphid can become viruliferous if it enters in contact with an infected plant. The probability per unit time that a non viruliferous aphid acquires the virus from an infected plant depends on i) the rate of contact between one aphid and a plant $\Lambda_i$, ii) the probability $\frac{I}{N_P}$ that the contact is indeed with an infected plant and iii) the probability $\varepsilon_i$ that the contact leads to virus acquisition. Viruliferous aphids lose viruliferousness at a clearance rate $\gamma$. We assume that the population of resident aphids varies following a logistic function with a density dependent *per capita* birth rate $r\left(1 - \frac{N_r}{h}\right)$, where $r$ is the intrinsic growth rate and $h$ is the plant hosting capacity with respect to aphid population (*i.e.* the resident aphid population size at which birth rate is zero), and a constant aphid mortality rate $\mu$. Resident aphids can leave their host plant and move to another one as a response to unfavourable environmental conditions [11]: we assumed $\Lambda_T > \Lambda_R > 0$, due to the different moving behaviour of transient and resident aphids. We assume that an average of $\lambda$ transient aphids per plant immigrate into the system (*i.e.* a patch of 1 ha) per unit time and a fraction $\pi$ of them are viruliferous. Such a fraction depends on the disease prevalence in the surrounding area. Transient aphids emigrate from the system or die at a rate $\tau$, which is the inverse of their average sojourn time in the system.

We assume that resident aphids exert two types of interference towards transient aphids, "visiting" and "emigration" interference, that will be presented in the following sections.

## Model equations

The model outlined above can be described by the following set of differential equations:

$$
\begin{cases}
\dot{S} = \Gamma - (\Lambda_R \delta_R Z_R + \Lambda_T \delta_T f(\cdot) Z_T)S - \theta S \\
\dot{I} = (\Lambda_R \delta_R Z_R + \Lambda_T \delta_T f(\cdot) Z_T)S - (\rho + \theta)I \\
\dot{X}_R = rN_R(1 - \dfrac{N_R}{h}) - \mu X_R - \Lambda_R \varepsilon_R \dfrac{I}{N_P} X_R + \gamma Z_R \\
\dot{Z}_R = \Lambda_R \varepsilon_R \dfrac{I}{N_P} X_R - (\gamma + \mu) Z_R \\
\dot{X}_T = (1 - \pi)\lambda - \Lambda_T \varepsilon_T f(\cdot) \dfrac{I}{N_P} X_T + \gamma Z_T - \tau g(\cdot) X_T \\
\dot{Z}_T = \pi\lambda + \Lambda_T \varepsilon_T f(\cdot) \dfrac{I}{N_P} X_T - \gamma Z_T - \tau g(\cdot) Z_T
\end{cases}
\tag{1}
$$

Where the dot represents the derivative with respect to time $t$, and parameters are as in Table 1. Functions $f(\cdot)$ and $g(\cdot)$ represent, respectively, "visiting interference" and "emigration interference" exerted by resident aphids towards transient aphids. Details on their functional forms are given in the following section.

The set of differential equations can be used to represent a plant virus epidemic under very general circumstances. To gain insight on disease transmission, we assume that the total number of plants ($N_P$) is constant (*i.e.* every harvested or rogued plant is immediately replaced), which implies that the number $\Gamma$ of new susceptible plant planted per unit time is given by

$$\Gamma = \theta S + (\rho + \theta)I \tag{2}$$

where $\theta$ is plant harvesting rate and $\rho$ is infected plant roguing rate. Moreover, the total number of transient aphids per plant in absence of resident aphids ($T$) is constant, which implies that the average number $\lambda$ of transient aphids immigrating into the system per plant per unit time is given by

$$\lambda = \tau T \tag{3}$$

## Modelling interference between resident and transient aphids

We assume that interference exerted by resident towards transient aphids can independently induce them to: *i*) visit fewer plants per unit time (visiting interference); and/or *ii*) reduce the average sojourn time in the system (emigration interference). Moreover, for both visiting and emigration interference, we consider two interference scenarios:

- direct (*e.g.* competition for space, [9]), where interference depends upon the density of resident aphids on the host ($\frac{N_R}{h}$). This implies that, at the same aphid abundance ($N_R$), the exerted interference is weaker on a bigger plant (*i.e.* higher plant hosting capacity $h$).

- Indirect (*e.g.* release of plant volatiles, [18]), where interference depends upon the absolute number of resident aphids and it is independent from the plant hosting capacity.

Agricultural practices, such as fertilization and irrigation, possibly influencing plant size, may have different effects on epidemic dynamics according to the interference scenario. Note that intermediate forms of interference, accounting for both competition for space and the release of plant volatiles, are possible and can be considered in our model by a proper parametrization (see S1 Text).

**Visiting interference.**   Visiting interference controls the proportionate decrease in the rate at which transient aphids visit plants, via two functional forms $f\left(\frac{N_R}{h}\right)$ and $f(N_R)$, respectively, for direct and indirect interference scenarios:

$$f\left(\frac{N_R}{h}\right) = \frac{1}{1 + \left(v_1 \dfrac{N_R}{h}\right)^{\alpha_1}} \quad f(N_R) = \frac{1}{1 + \left(\dfrac{v_1}{h_R} N_R\right)^{\alpha_1}} \tag{4}$$

These are a generalisation of the competition function proposed by Bellows [41, 42] for insect populations, extending in continuous time the model of Maynard Smith and Slatkin [43]. They are sufficiently flexible to account for a range of possible types of interference (Fig 2). The "strength" parameter $v_1$ controls the magnitude of interference, and so the density of residents that is required to appreciably affect the behaviour of transient aphids (*i.e.* stronger interference for higher values of $v_1$). In our model the value of $v_1$ is defined in reference to the direct interference scenario. To assure that its value is biologically relevant also in the indirect

**Table 1. Model state variables and parameters.**

| Variable | Description | Dimensions | | |
|---|---|---|---|---|
| $S$ | Susceptible plants | plant ha$^{-1}$ | | |
| $I$ | Infected plants | plant ha$^{-1}$ | | |
| $X_R$ | Non viruliferous resident aphids | aphid plant$^{-1}$ | | |
| $Z_R$ | Viruliferous resident aphids | aphid plant$^{-1}$ | | |
| $X_T$ | Non viruliferous transient aphids | aphid plant$^{-1}$ | | |
| $Z_T$ | Viruliferous transient aphids | aphid plant$^{-1}$ | | |
| **Parameter** | **Description** | **Dimensions** | **Values** | **Source** |
| $\Gamma$ | Planting rate | plant day$^{-1}$ | $\theta S + (\rho + \theta)I$ | |
| $\Lambda_R$ | Number of plants visited by a resident aphid | plant aphid$^{-1}$ day$^{-1}$ | 0.05 | Fixed |
| $\Lambda_T$ | Number of plants visited by a transient aphid | plant aphid$^{-1}$ day$^{-1}$ | 8.5 | Fixed |
| $\delta_R$ | Probability of virus transmission from the resident aphid to the plant | dimensionless | 0.04 | [34] |
| $\delta_T$ | Probability of virus transmission from the transient aphid to the plant | dimensionless | 0.04 | [34] |
| $\varepsilon_R$ | Probability of virus transmission from the plant to the resident aphid | dimensionless | 0.02 | [34] |
| $\varepsilon_T$ | Probability of virus transmission from the plant to the transient aphid | dimensionless | 0.02 | [34] |
| $\alpha_1$ | Visiting interference curvature | dimensionless | 1.00 | Fixed |
| $v_1$ | Visiting interference strength (for direct interference) | dimensionless | 12.0 | Fixed |
| $\alpha_2$ | Emigration interference curvature | dimensionless | 1.00 | Fixed |
| $v_2$ | Emigration interference strength (for direct interference) | dimensionless | 12.0 | Fixed |
| $\rho$ | Infected plant roguing rate | day$^{-1}$ | 0.02 | [35, 36] |
| $\theta$ | Plant harvesting rate | day$^{-1}$ | 0.003 | [37] |
| $r$ | Intrinsic growth rate of resident aphids | day$^{-1}$ | 0.21 | [37] |
| $h$ | Plant hosting capacity | aphid plant$^{-1}$ | 50,000 | [38] |
| $h_R$ | Reference plant hosting capacity | aphid plant$^{-1}$ | 50,000 | Fixed |
| $\mu$ | Mortality rate of resident aphids | day$^{-1}$ | 0.08 | Fixed |
| $\gamma$ | Virus clearance rate in aphid vectors | day$^{-1}$ | 4 | [34] |
| $\lambda$ | Average number of transient aphids immigrating per plant | aphid plant$^{-1}$ day$^{-1}$ | $\tau T = 250$ | Derived |
| $\pi$ | Fraction of viruliferous transient aphids entering the system | dimensionless | 0 | Fixed |
| $\tau$ | Transient aphids emigration rate in absence of resident aphids | day$^{-1}$ | 0.5 | [35, 39] |
| $N_P$ | Total number of plants | plant ha$^{-1}$ | 720 | [40] |
| $T$ | Average number of transient aphids per plant in absence of resident aphids | aphid plant$^{-1}$ | 500 | Fixed |

Source: Fixed: fixed to an arbitrary, biologically-plausible reference value.

interference scenario, we scale it with the "reference" value of plant hosting capacity ($h_R$). This implies that, when $h = h_R$ and all other things being equal (*i.e.* the values of $\alpha_1$ and $N_R$), the value of the interference function is the same independently from the underlying interference scenario. The "curvature" parameter $\alpha_1$, controls whether the visiting function presents an inflection point (for $\alpha_1 > 1$) or not (for $0 < \alpha_1 \leq 1$): in the first case visiting interference starts playing a role (i.e. $f(\cdot) < 1$) once the population of resident aphids reaches a threshold value, in the second case it is always active for $N_R > 0$ (see Fig 2).

**Emigration interference.** Emigration interference controls the proportionate increase in the rate at which transient aphids leave the system, via two functional form $g\left(\frac{N_R}{h}\right)$ and $g(N_R)$, respectively for direct and indirect interference scenarios:

$$g\left(\frac{N_R}{h}\right) = 1 + \left(v_2 \frac{N_R}{h}\right)^{\alpha_2} \quad g(N_R) = 1 + \left(\frac{v_2}{h_R} N_R\right)^{\alpha_2} \tag{5}$$

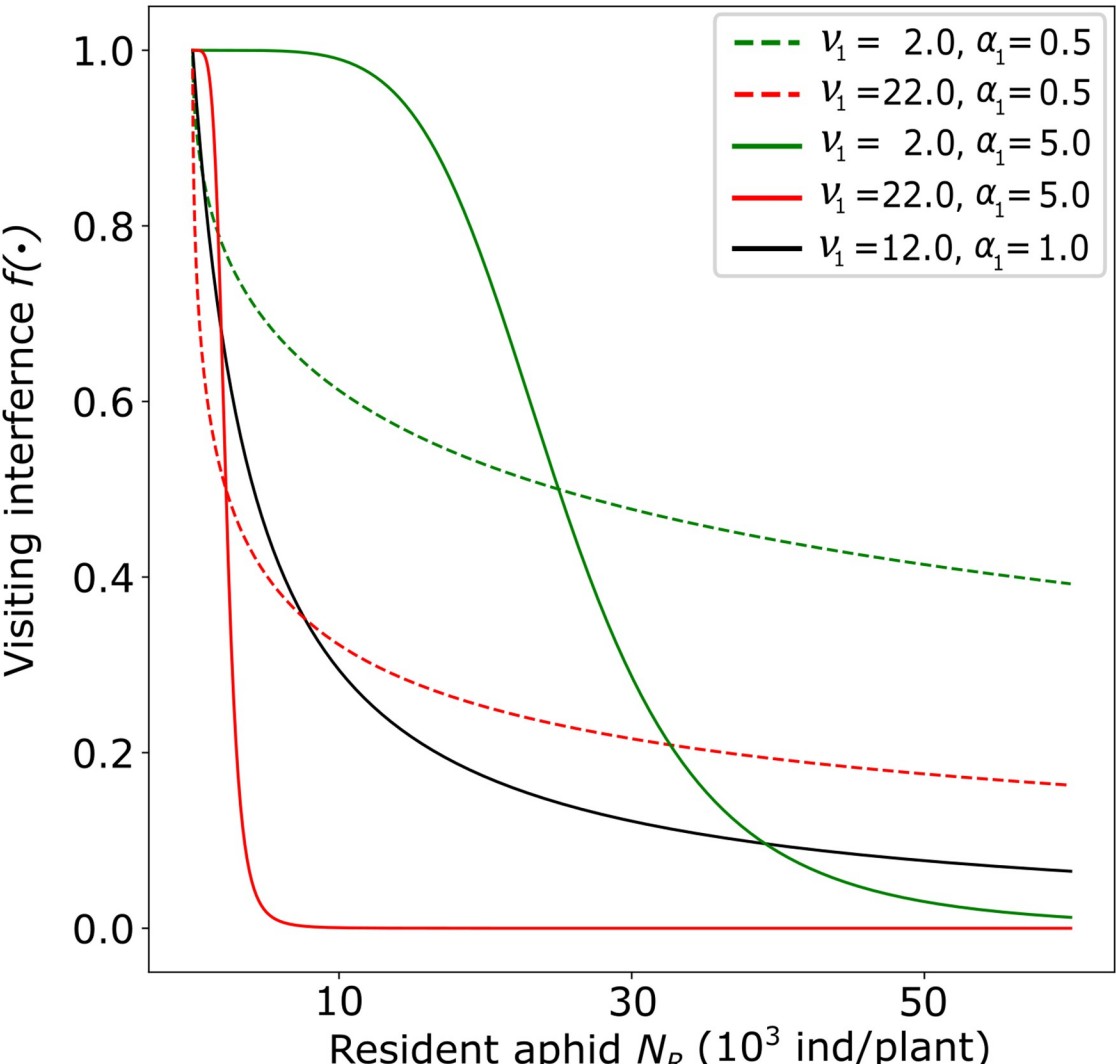

**Fig 2. Visiting interference function.** Proportionate decrease of the number of plants visited by a transient aphid as a function of the abundance of resident aphids, for three values of parameters $v_1$ and $\alpha_1$, in a plant with a reference plant hosting capacity (*i.e.* $h = h_R$, thus the value of $f(\cdot)$ is independent of the interference scenario). The black curve represents the visiting interference function for the parameter values used in numerical simulations.

The emigration interference function is characterized by two parameters determining strength and curvature ($v_2$ and $\alpha_2$), and, in case of indirect interference scenario, also by the parameter $h_R$. All three parameters have identical interpretations to those controlling the visiting interference. In the most general form of our model, the analogous parameters for the two functions are entirely independent (*e.g.* $\alpha_1 \neq \alpha_2$), which provides sufficient flexibility to allow inter-specific interference to affect visiting and/or emigration independently. However in all numerical simulations in this paper we assumed that $v_1 = v_2 = v$ and $\alpha_1 = \alpha_2 = \alpha$, and so, in turn, that proportionate effects of the density of resident aphids on visiting and emigration rate were similar.

## Agricultural practices and disease control

A number of agricultural practices are commonly used to *i*) increase plant growth, *ii*) control aphid populations and *iii*) mitigate the effect of plant diseases. The effect of these practices can be taken into account by modifying the values of some model parameters from their reference value. Practices such as fertilization or irrigation, commonly used to foster crop growth, may increase the abundance of resident aphids dwelling on the plants [44, 45] and in our model this translates into an increase of the plant hosting capacity $h$ (while parameter $h_R$ is not varied). Pesticides are commonly used in crops to reduce the number of resident aphids and in our model this translates into an increase of resident aphid mortality $\mu$, while the dynamics of transient aphids is not affected [46]. In case of spread of plant diseases, frequently producers try to identify as soon as possible the infected plants and replace them with new healthy plants, a practice known as roguing [47]. To represent the discrete time process of roguing in our continuous time model, we set the rate of removal of infected host plants ($\rho$) such that the average period for which plants are symptomatically infected is equal to one half of the interval between successive rounds of roguing [21].

## Methods of analysis

We first determine the system equilibria and the basic reproduction number of the disease, $R_0$. Then, we explore the responses of $R_0$ to variations of the aforementioned agricultural practices (modeled through parameters $h$, $\mu$ and $\rho$) under the direct and indirect interference scenarios. In particular, we vary one parameter at a time and we explore the effectiveness of the respective agricultural practices in controlling and eradicating the disease. Finally, we analyse the effectiveness of combinations of agricultural practices in controlling and eradicating the disease by exploring the response of $R_0$ to simultaneous variation of two parameters at a time. In our analysis, we use biologically plausible parameter values and ranges to reflect a broad range of single host-multi vector system rather than restricting the analysis to a specific system.

# Results

## Equilibrium analysis

The behaviour of the model at equilibrium is summarized in Table 2. When $\mu \geq r$ (*i.e.* the mortality rate of resident aphids is larger than their intrinsic growth rate), resident aphids are not able to survive ($\bar{X}_R = \bar{Z}_R = 0$) and the disease is spread exclusively by transient aphids. When $\mu < r$ both resident and transient aphids are present in the system and may spread the disease. When all the incoming transient aphids are non viruliferous (*i.e.* $\pi = 0$), the disease is able to persist only if the basic reproduction number ($R_0$, presented in the following section) is higher than 1, otherwise the disease is eradicated [48]. When a fraction of incoming transient aphids is viruliferous ($\pi > 0$), the disease is always able to persist because there will always be an influx of some new viruliferous aphids into the system, and infections of plants will result not just from infected plants in the system, but also from viruliferous individuals originating from outside the system (similar to [39]). In this case, it is not possible to define $R_0$ as a threshold of disease persistence. The patterns presented in Table 2 can be explained analytically, with the mathematical details derived in S2 Text.

**Table 2. Summary of equilibrium behaviour.** The value of state variables at the equilibrium are presented in Table A in S2 Text.

| Viruliferous aphids enter the system ($\pi > 0$) | Resident aphids are present ($\mu < r$) | Basic reproduction number (Eq 6) | $(\bar{S}, \bar{I}, \bar{X}_R, \bar{Z}_R, \bar{X}_T, \bar{Z}_T)$ | Explanation |
|---|---|---|---|---|
| no | no | $R_0 < 1$ | $(+, 0, 0, 0, +, 0)$ | Transient aphids do not bear the disease from outside the system. Resident aphids are absent, the disease is spread by transient aphids but it does not persist in the system. |
| no | no | $R_0 > 1$ | $(+, +, 0, 0, +, +)$ | Transient aphids do not bear the disease from outside the system. Resident aphids are absent, the disease is spread by transient aphids. |
| no | yes | $R_0 < 1$ | $(+, 0, +, 0, +, 0)$ | Transient aphids do not bear the disease from outside the system. Resident and transient aphids spread the disease, but it does not persist in the system. |
| no | yes | $R_0 > 1$ | $(+, +, +, +, +, +)$ | Transient aphids do not bear the disease from outside the system. Resident and transient aphids spread the disease. |
| yes | no | - * | $(+, +, 0, 0, +, +)$ | Transient aphids bear the disease from outside the system. Resident aphids are absent, transient aphids spread the disease. |
| yes | no | - * | $(+, +, +, +, +, +)$ | Transient aphids bear the disease from outside the system. Resident and transient aphids spread the disease. |

* The disease is always able to persist, the basic reproduction number is not definable.

## The basic reproduction number

In our system, when there is no immigration of viruliferous aphids ($\pi = 0$), the basic reproduction number $R_0$ is expressed as

$$R_0 = \frac{1}{\rho + \theta}\left(\frac{\Lambda_R^2 \delta_R \varepsilon_R \bar{N}_R}{\gamma + \mu} + \frac{\Lambda_T^2 \delta_T \varepsilon_T f(\cdot)^2 \bar{N}_T}{\gamma + \tau g(\cdot)}\right) \qquad (6)$$

Here we follow common practice in plant disease epidemiology [39, 49] and use $R_0$ to refer to the threshold quantity as obtained by heuristic interpretation of the terms in our differential equation model. Strictly-speaking, the quantity identified in Eq 6 is actually $R_0^2$ for the system, since two cycles are involved in transmission, *i.e.* from plant to vector and from vector to plant [50]. However, since the two thresholds predict identical behaviour in terms of disease invasion (the threshold $R_0 = 1$ is precisely equivalent to $R_0^2 = 1$), we prefer to use the simpler formulation here.

Note that our basic reproduction number can be written as the sum of two components, $R_0 = R_0^R + R_0^T$, where $R_0^R = \frac{\Lambda_R^2 \delta_R \varepsilon_R \bar{N}_R}{(\rho+\theta)(\gamma+\mu)}$ and $R_0^T = \frac{\Lambda_T^2 \delta_T \varepsilon_T f(\cdot)^2 \bar{N}_T}{(\rho+\theta)(\gamma+\tau g(\cdot))}$ (see S2 Text for further details). The first component accounts for disease transmission by resident aphids and the second by transient aphids. Such a representation of the basic reproduction number is typical for plant disease models with multiple routes of transmission [51–53]. In the equation of the basic reproduction number, the term $(\rho + \theta)^{-1}$ indicates the average time spent by a plant as infected, before it is rogued or harvested; the terms $(\gamma + \mu)^{-1}$ and $(\gamma + \tau g(\cdot))^{-1}$ indicate the average time spent by a viruliferous aphid, respectively resident or transient, in the system. The terms $\Lambda_R \delta_R \bar{N}_R$ and $\Lambda_T \delta_T f(\cdot)\bar{N}_T$ represent the rates at which a susceptible plant is infected by resident or transient aphids, respectively. The terms $\bar{N}_R$ and $\bar{N}_T$ indicate the number of, respectively, resident and transient aphids per plant at the steady state. The terms $\Lambda_R \varepsilon_R$ and $\Lambda_T \varepsilon_T f(\cdot)$ are the rates at which a resident or a transient aphid acquires the virus from an infected plant, respectively. The terms $f(\cdot)$ and $g(\cdot)$ are, respectively, the function for visiting and emigration interference evaluated at the steady state, for the direct interference scenario

$[f(\cdot) = f\left(\frac{\bar{N}_R}{h}\right)$ and $g(\cdot) = g\left(\frac{\bar{N}_R}{h}\right)]$ and for the indirect interference scenarios $(f(\cdot) = f(\bar{N}_R)$ and $g(\cdot) = g(\bar{N}_R))]$.

The size of the resident aphid population $(\bar{N}_R)$ appears both at the numerator of the $R_0^R$ component and as an argument of the interference functions $f(\cdot)$ and $g(\cdot)$ which respectively decrease and increase with $\bar{N}_R$. This implies that the resident component, $R_0^R$, of the basic reproduction number increases with $\bar{N}_R$ while the transient component, $R_0^T$, decreases with it. According to the assumed population dynamics of resident and transient aphids, $\bar{N}_R = h(1 - \mu/r)$ if $\mu < r$, $\bar{N}_R = 0$ if $\mu \geq r$ and $\bar{N}_T = \lambda/(\tau g(\cdot))$ (see S2 Text for further details). This suggests that the response of the basic reproduction number to $h$, $\mu$ and $r$ might be non-monotonic.

Note that, whenever the basic reproduction number is higher than 1, the disease is not erad-icated and the incidence of virus infection in plant can be computed as $\frac{\bar{I}}{N_P}$, where $\bar{I}$ is the size of infected plant population at equilibrium and $N_P$ is the total number of plants in the system (assumed constant). The response of the incidence of virus infection in plants to variation of parameters $h$, $\mu$ and $\rho$ correlates strongly with that of $R_0$ (see S3 Text).

## The role of plant hosting capacity

The response of the basic reproduction number to aphid hosting capacity is summarized in Fig 3A. When the system conditions can sustain a population of resident aphids (*i.e.* $r > \mu$), its equilibrium value increases with the plant hosting capacity $h$. This always translates into an increase of the term $R_0^R$. On the other hand, the effect of increasing $h$ on the term $R_0^T$ is medi-ated by the interference functions, $f(\cdot)$ and $g(\cdot)$. Such an effect is null when the interference is direct (see Eqs 4 and 5, for $\bar{N}_R = h(1 - \mu/r)$) and it is negative when the interference between aphids is indirect. Increasing plant size, and consequently its hosting capacity, is followed by an increase of the population of resident aphids dwelling on a plant. This has no effect on the transient aphid population when the interference is direct, because the density of resident aphids is kept constant, but it negatively affects the transient aphid population when the inter-ference is indirect. When the interference is direct, increasing the value of $h$ increases the basic

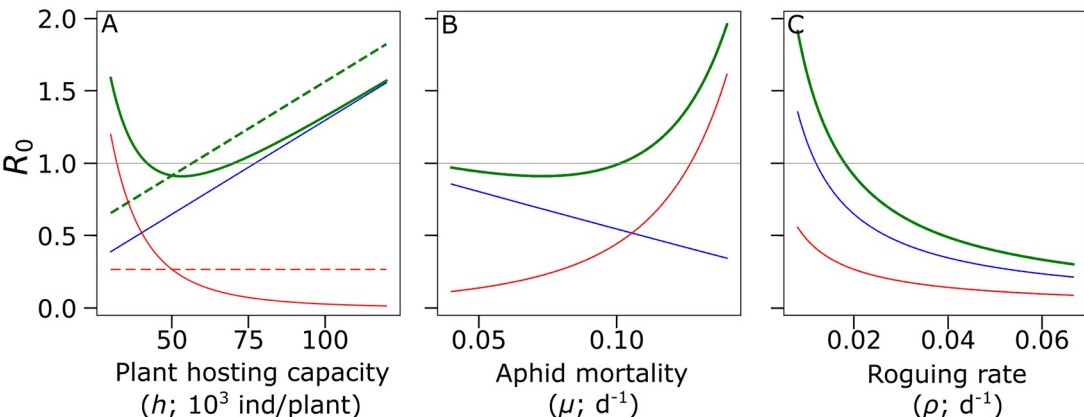

**Fig 3. Effects of agricultural practices on the basic reproduction number.** Response of the basic reproduction number $R_0$ (in bold and green) and its components $R_0^R$ (in blue) and $R_0^T$ (in red) to changes in (A) plant hosting capacity ($h$) under indirect (continuous line) and direct (dashed line) interference scenarios, (B) resident aphids mortality ($\mu$), (C) roguing rate ($\rho$). Note that in (A) blue continuous and dashed lines overlap.

reproduction number but, in the presence of indirect interference the response of $R_0$ is non-monotone, with a minimum value obtained for intermediate values of hosting capacity.

## The role of resident aphid mortality

The effect of aphid mortality ($\mu$) on the basic reproduction number is summarized in Fig 3B. Increasing $\mu$ has a negative effect on the $R_0^R$ term reducing both the resident population density and the average time spent by a viruliferous resident aphid in the system. On the other hand, it has a positive effect on the $R_0^T$ term by releasing interference forces exerted by resident aphids and consequently increasing transient aphids movement and sojourn time in the system. In this case, the interference scenario (whether direct or indirect) has no influence because the plant hosting capacity is considered at its reference value $h = h_R$ (see Eqs 4 and 5, for $h = h_R$). Also in this case, a minimum value of $R_0$ is obtained for intermediate values of resident aphid mortality.

## The role of roguing

An increase to the roguing rate $\rho$ decreases the basic reproduction number as it reduces the time that an infected plant spends in the system before it is rogued (see Eq 6 and Fig 3C). As before, the interference scenario has no influence on the value of $R_0$ because the plant hosting capacity is considered at its reference value $h = h_R$.

## Agricultural practices and disease control

The response of $R_0$ to the simultaneous variation of two control parameters is summarized in Fig 4. It is always possible to eradicate the disease for some combination of two of the considered parameters. Eradication is possible without the addition of pesticides ($\mu = 0.04$ day$^{-1}$, *i.e.* resident aphid natural mortality) for relatively small plant hosting capacity ($h$, which increases with fertilization and irrigation) (Fig 4A and 4B). Eradication is not possible for relative high values of resident aphid mortality ($\mu$), independently from the value of $h$, in the direct interference scenario (Fig 4B), while it is possible in the indirect interference scenario, where the value of $\mu$ that leads to disease eradication, increases with the value of $h$ (Fig 4A). Increasing roguing rate ($\rho$) increases the range of $h$ and $\mu$ values that lead to disease eradication. Both for direct and indirect interference scenarios, if infected plant are identified and eliminated every periods of a maximum of 50 days (*i.e.* $\rho \geq 0.04$ day$^{-1}$), disease eradication is possible for nearly all the considered $h$ values (Fig 4C and 4D) and for nearly all the considered $\mu$ values (Fig 4E).

## Discussion

Meta-analytical studies [14, 15] suggests that both direct (*i.e.* aphid mediated) and indirect (*i.e.* plant mediated) interferences between herbivorous insects shape their behaviour and performance. For example, in an experimental work, Mehrparvar and colleagues [17] showed that interference between different aphids species affects host selection behaviour, with aphid individuals rarely choosing a plant occupied by individuals of another aphids' species. However, these interference mechanisms have been ignored in epidemiological analyses despite a few exceptions (*e.g.* [16, 32, 54]). Yet, to our knowledge, the existing experimental and theoretical works only consider interferences between vector and non-vector insects. For example, Chisholm and colleagues [16] observed higher rates of *Pea enation mosaic virus* spread when the vector *Acythosiphon pisum* individuals shared hosts with a non vector herbivore *Sitona lineatus*.

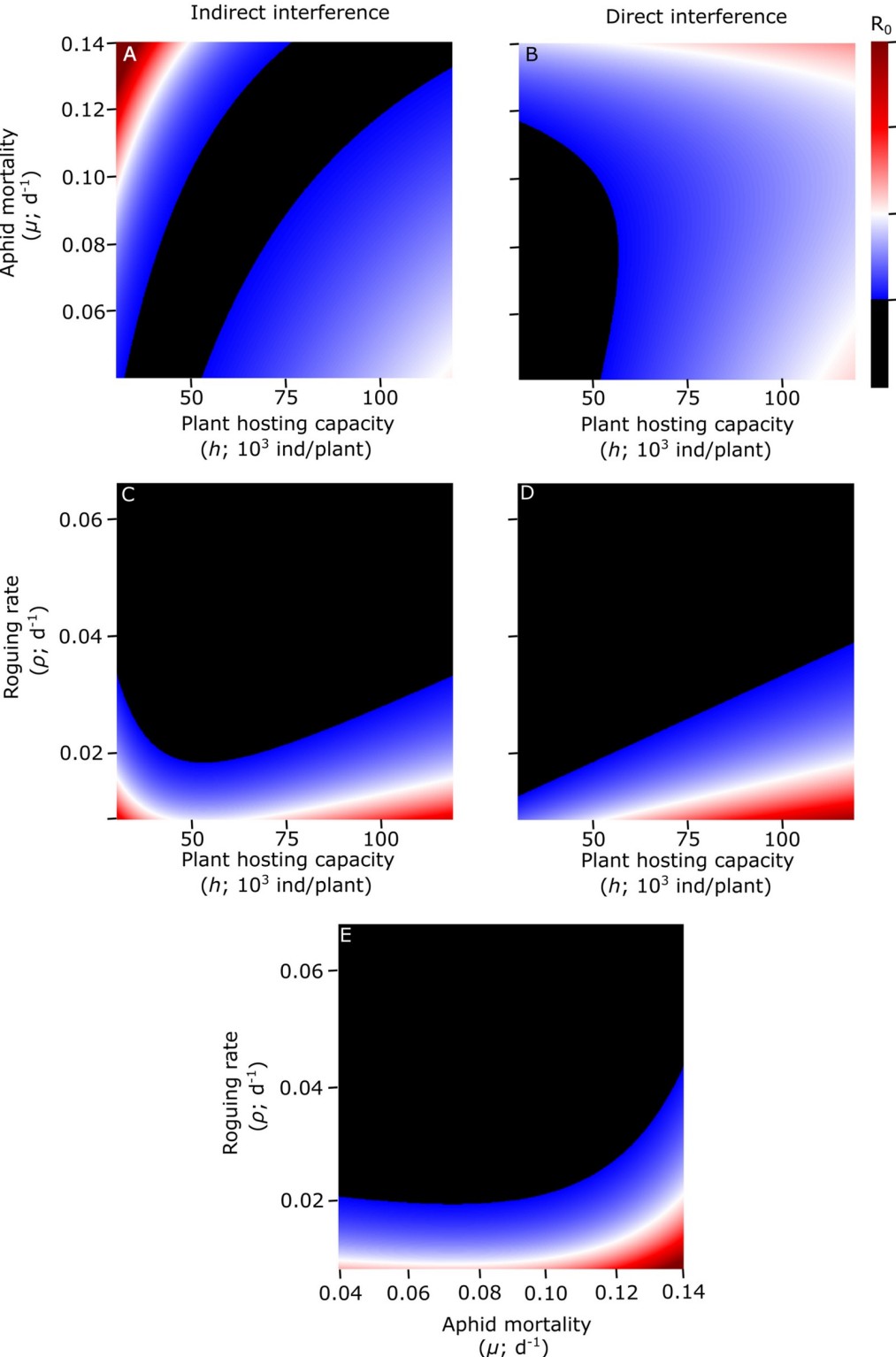

**Fig 4. Effects of combinations of agricultural practices on the basic reproduction number.** Response of $R_0$ to changes in: plant hosting capacity ($h$) and resident aphid mortality ($\mu$) (A-B); plant hosting capacity ($h$) and roguing rate ($\rho$) (C-D); resident aphid mortality ($\mu$) and roguing rate ($\rho$) (E), under different interference scenarios (indirect and direct). Note that the interference scenario has no effect on $R_0$ when $\mu$ and $\rho$ are simultaneously varied (E). Black areas identify values of $R_0 < 1$, corresponding to disease eradication. Other model parameters are set to default values (Table 1).

In the present work, we modeled the interaction between two aphid vectors via two interference functions, which account for the reduction of the number of plants visited per unit time and the average resident time in the system of transient aphids. While it is crucial to have chosen monotonic interference functions, the precise functional forms and values of parameters are not particularly important to set the responses we showed, which are qualitatively unaffected by the values of the parameters $\nu$ and $\alpha$ of the interference functions (see S4 Text). Both direct and indirect interference scenario can be represented depending on the considered pathosystem. By means of mathematical and numerical analyses of our model, we have demonstrated that interference can have profound effects on the invasion, persistence and control of plant NPT viruses. Other theoretical works have shown that altering vector movement, in a way that limits the number of plants visited, might be more effective in controlling NPT viruses rather than reducing vector abundance in the system [32]. Indeed, when the length of time an infectious vector spends moving, searching for a new host to visit, is long in comparison to pathogen retention time, the pathogen may be cleared in transit before visiting a new susceptible host [3, 55]. The theoretical analysis of our model confirms these findings: visiting interference $f(\cdot)$, affecting the number of plants visited by transient aphids, reduces both the probability of acquiring and inoculating the virus. That is why, no matter whether the underlying mechanism is direct or indirect, visiting interference appears squared in the basic reproduction number, which determines a greater effect of the visiting interference in diminishing the invasion and persistence of the disease respect to emigration interference.

We used a minimum number of epidemiological compartments (*i.e.* $S$ and $I$) for the sake of simplicity and to minimize the number of parameters. Considering a compartment for exposed $E$ (also known as latently infected) individuals would make no sense for vectors of NPT viruses: virus particles are attached to the insect's stylets and rapidly inoculated into plant, making the insect immediately infectious after virus acquisition [4, 56]. Although some plants may spend some time in an exposed, yet not infectious state, this would not have a qualitative impact on our results regarding the basic reproduction number. In that case, a multiplicative term $\frac{\omega}{\omega + \theta}$ will be added to the $R_0$ formula (where $\omega$ is the inverse of the plant latent period and $\theta$ is the plant harvesting rate). This term represents the probability of an exposed plant becoming infectious, which is close to 1 for small value of $\theta$. Furthermore, the presence of an exposed compartment would affect the rate of disease increase similarly to other parameters (*e.g.* probability of virus transmission from plant to aphids and vice versa, virus clearance rate) already considered in our model [57]. A compartment of recovered ($R$) individuals makes little sense for vectors of NPT viruses which continuously shift between a viruliferous and non-viruliferous state. Also, for plants we assumed that infected plants do not recover as commonly assumed for viral plant diseases [58]. By considering roguing and harvesting in our model, we showed what would happen if a removed compartment is included.

Our results suggest that commonly used agricultural practices, such as fertilization and irrigation, pesticide application and roguing of diseased plants, can have unexpected results upon the spread of NPT viruses. Assuming that all the immigrants transient aphids are non infectious ($\pi = 0$) allowed us to focus on the variations of the basic reproduction number ($R_0$), which we have defined in terms of the square root of the threshold as obtained via the so-called Next Generation Matrix [59]. This methodology, which allows relatively simple calculation of the basic reproduction number, requires as a first step the linearization of the system around its disease-free-equilibrium. As shown in our mathematical analysis, in the case for $\pi > 0$, the disease is always able to persist. Since there is no disease-free-equilibrium, it is not possible to define $R_0$ when $\pi > 0$. Of course one could assume, as it often occurs in real systems [60], that a transient vector carries NPT viruses from neighbouring crops. In this case, our modelling

framework would permit to disentangle the role of resident and transient vectors and simulate the consequences of agronomic practices but an eradication of the disease would not be possible.

Fertilization and irrigation are commonly used in agriculture to meet plants' nutrient and water needs and increase plant growth and production [61]. Yet they can impact disease development and spread, possibly affecting plant physiology, pathogens and/or vector population dynamics [62, 63]. On the one hand nitrogen is involved in the resistance mechanisms of the host plant, *i.e.* its ability to limit the development and reproduction of the invading pathogen, possibly decreasing the incidence of disease in crop plants [62]. On the other hand, fertilization and irrigation can increase vector populations via changes in plant nutrient and irrigation status, potentially impacting the spread of plant diseases. For example, the growth rate of wheat curl mite, vector of the *Wheat streak mosaic virus*, was observed to increase with fertilization on winter wheat [63]. Populations of bird cherry-oat aphid (*Rhopalosiphum padi L.*), vector of the *Barley yellow dwarf virus*, have been observed to increase with irrigation [64] and fertilization [65] on different grass species. Our results show that an intermediate plant size, which sustains a population of resident aphids large enough to appreciably reduce the spread of the virus by transient aphids, but not too large to prevent disease spread by resident vectors, may lead to disease eradication.

Pesticide application is the most common aphid control method, but it is well known that its ability to prevent the spread of NPT viruses by transient aphids is limited because inoculation occurs rapidly and before a pesticide can take effect on the transient vector [46]. Transient aphids rapidly pass through the field, landing on plant and assessing it through epidermic exploratory probing. After such exploratory probing, they quickly disperse to another host, having rejected the plant as unpalatable. Accounting for the short time transient aphids spend on plants, it is difficult to imagine that they are exposed to a lethal dose of pesticide before leaving the field [46, 66]. Only pesticides that impact the initial stages of plant discovery and assessment before exploratory probing may reduce NPT viruses transmission [46]. For example, pesticides which repel the vector are likely to be useful in controlling the spread of NPT viruses [46]. This is similar to transient aphids repulsion due to plant volatiles included in our work by mean of the indirect interference scenario. Furthermore pesticide may affect local pest community structure as differential susceptibility to pesticide may result in species dominance shift favoring secondary pest outbreaks [67, 68]. In their experiments with two aphids species, *Rhopalosiphum padi* and *Sitobion avenae*, Mohammed and colleagues [67] showed that pesticide exposure led to a shift in the outcome of interspecific competition between the two aphid species, compromising the dominance of *R. padi* in pesticide-free plants, while favouring the prevalence of the *S. avenae* under pesticides exposure. Our results show that small pesticide application has the potential to slightly reduce the spread of NPT viruses. However, large pesticide application, reducing the interference exerted by resident towards transient aphids, could be counter productive in reducing NPT viruses, because it favours the prevalence of transient aphids, increasing the spread of the virus by the more mobile vector.

Roguing infected plants has often been implemented to control the spread of plant pathogens [6, 47]. The success of roguing in slowing disease spread depends on how rapidly infected plants are identified and removed [69, 70]. Yet, there are various logistical issues associated with identification and removal of infected plants in large-scale agriculture. Firstly, the identification of diseased plants may be hampered by a lack of appropriate and/or cost-effective diagnostic tests. Further, growers can be reluctant to remove diseased plants as soon as symptoms are identified, since infected plants may continue to produce a marketable yield [47]. Finally, the degree of coordination among farmers concerning the decision of roguing is likely to affect the success in slowing disease spread [71]. Our results unsurprisingly suggest that the

incidence of the disease decreases with the effort put into roguing. Yet, they would benefit from further economic evaluations, given the cost of roguing and replanting operations. We assumed that removed plants are replaced by healthy one as commonly done by a grower that wants to maintain a certain production per hectare. If it were not the case, this would lead to a non constant host population and further analyses should be done to assess the robustness of our findings.

Despite our efforts to provide a realistic representation of the complex epidemiological and ecological components of an agroecosystem, we had to introduce a number of simplifying assumptions. We have assumed the resident aphid mortality rate to be constant, in accordance to several authors [30–32]. In reality, the effects of chemical control on pest mortality do not remain constant, but vary with the repeated application and subsequent decay of pesticides' concentration. We have assumed that NPT viruses do not manipulate host-vector behavior in order to enhance their own transmission, for example by making infected plant more attractive to aphids but inhibiting aphid settling on infected plants [31, 72]. This may not always hold, for example, it was shown that squash plants (*Cucurbita pepo*) infected with *Cucumber mosaic virus* firstly emit a blend of volatile organic compounds that attracts aphids, and secondly produce anti-feedant compound, which deter aphids from prolonged feeding [73]. Yet, a non negligible number of pathosystems involving viruses considered to be of the NPT transmission type do not follow this "attract and deter" trend [72]. Finally, it is possible that plants put in place other types of defensive mechanisms which may impair resident aphid fecundity [18] and which can be fostered by fertilization [74]. Although all these mechanisms could be included in our model, we have chosen to avoid the proliferation of parameters which would have been associated with more complex models, possibly hiding the underlying message of this work. Yet, despite the simplifying assumptions outlined above and noting that further experimental works are clearly required to confirm our findings, our work suggests that the impacts of inter-specific interference should be incorporated more broadly into the planning of disease management strategies for the control and eradication of aphid vectored NPT viruses.

## Supporting information

**S1 Text. Generalized interference functions.**
(PDF)

**S2 Text. Mathematical analysis of the single host-multi vector model.**
(PDF)

**S3 Text. Agricultural practices and disease control.**
(PDF)

**S4 Text. Influence of interference parameters on results.**
(PDF)

## Author Contributions

**Conceptualization:** Marta Zaffaroni, Loup Rimbaud, Nik J. Cunniffe, Daniele Bevacqua.

**Formal analysis:** Marta Zaffaroni, Nik J. Cunniffe.

**Funding acquisition:** Daniele Bevacqua.

**Investigation:** Marta Zaffaroni.

**Methodology:** Marta Zaffaroni, Nik J. Cunniffe, Daniele Bevacqua.

**Project administration:** Daniele Bevacqua.

**Software:** Marta Zaffaroni.

**Supervision:** Ludovic Mailleret, Nik J. Cunniffe, Daniele Bevacqua.

**Validation:** Marta Zaffaroni.

**Visualization:** Marta Zaffaroni.

**Writing – original draft:** Marta Zaffaroni.

**Writing – review & editing:** Marta Zaffaroni, Loup Rimbaud, Ludovic Mailleret, Nik J. Cunniffe, Daniele Bevacqua.

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
