## [Decision Letter · Decision Letter 0]

28 Oct 2021

Dear Dr. Cunniffe,

Thank you very much for submitting your manuscript "The role of interference between vectors in control of plant diseases" for consideration at PLOS Computational Biology. As with all papers reviewed by the journal, your manuscript was reviewed by members of the editorial board and by several independent reviewers. The reviewers found the paper interesting, while suggesting a few areas where minor changes could improve the presentation. Based on the reviews, we are likely to accept this manuscript for publication, providing that you modify the manuscript according to the review recommendations.

Sincerely,

Konstantin B. Blyuss

Guest Editor

PLOS Computational Biology

Virginia Pitzer

Deputy Editor-in-Chief

PLOS Computational Biology

[LINK]

Reviewer's Responses to Questions

**Comments to the Authors:**

Reviewer #1: Plant viruses are major constraints to crop production worldwide. Most plant viruses are transmitted by insect or arthropod vectors, with several different transmission mechanisms. These transmission mechanisms lead to virus groupings that are not based on virus taxonomy, per se, but on epidemiologically relevant traits (time to inoculate a plant, time for vector to acquire the virus from a plant, infectious period in the vector, transmission to insect offspring, and so on). Nonpersistently transmitted (NPT) viruses comprise a large group with many economically and ecologically important viruses, many of which are transmitted by aphids (which can live/reside within the crop or are transitory, passing through the crop rather quickly while also transmitting a virus). There can be interactions between the resident and transient vector populations (direct and indirect), especially interference of transients by the residents, the subjects of this manuscript.

Although there have been several strong theoretical studies on modeling plant virus disease epidemics (properly cited in this manuscript), this is the first major theoretical modeling effort aimed at the impact of interference of vectors on the plant virus epidemics. Building on past theoretical and empirical work, the authors developed an extended deterministic SI coupled differential equation type model for plant viruses and their vectors. They define the many parameters and variables, and explore the impacts of vector interference AND some cropping systems conditions (fertilization, irrigation) on the plant epidemics. Importantly, they derived R0 for a subset of the conditions covered by the SI model. Although the model is (necessarily?) quite complex, the components are explained and justified, and they appear to be based on the current knowledge of the epidemiology of NPT plant viruses. The authors show the situations that lead to epidemics and no epidemics based on R0 (and the parameters that make up R0).

The manuscript does not present any empirical data, and I doubt that the SI model could ever be fit directly to data without pre-assigning most of the parameters to fixed values (or exploring different [highly] informative priors for them using Bayesian methods). However, I think there are sufficient interesting and illuminating theoretical results to justify acceptance. I found the results very interesting. These modeling results can inspire future field research. Results, as expected, show that there may a complex relation between epidemiological (or agronomic) parameters and epidemic outcomes. Vector interference is a epidemiological process that needs to be considered for developing control strategies for plant viruses with NPT vectors.

Although I feel the manuscript is valuable and warrants acceptance, I think the authors should consider a few items in their revision, as explained below. These are mostly for the Discussion.

Title: Since this manuscript is for NPT viruses only, I think the title should reflect this. (The epidemiological properties of semi-persistently and persistently transmitted plant viruses are different). I also think that the title should reflect that this is a theoretical study.

The f(.) and g(.) equations 4 and 5 should have greater justification or explanation in the main body of the text (not just the supplement). I would like greater justification for the fixed parameters that are chosen, and a better explanation of the shape of the curves that are produced.

The authors clearly explain that their derived R0 (eqn 6) is only for the situation when there are no immigrant viruliferous (infectious) vectors (pi=0). In fact, I think that R0 may not be definable here is pi>0), although I would have to think further about that. The authors nicely explore the effects of many important epidemiological parameters on R0 in this manuscript. But what can we conclude about the situation (and especially the role of interference) when pi>0 in terms of plant virus epidemics? A not uncommon scenario is for immigrant vectors to be carrying the virus (say, from surrounding infected weeds or other neighboring crops). Many years ago, Mike Irwin studied this in the field with soybean NPT viruses. This is probably a new theoretical investigation for the future, but the authors should certainly address this more in the Discussion.

The authors have taken a strictly SI modeling approach. Although this is adequate for their objectives (in my view), they should discuss in the Discussion some of the implications of this simplifying modeling choice (of course, the model is complex in other ways). For instance, they ignore the latent (E, exposed) and removed (R) diseased states in plants. This does create a more artificial situation (or very specific) epidemiological situation than found in general for plant epidemics (although a common simplifying SEIR modeling tool). Ignoring E (i.e., assuming that the latent period is 0) won’t effect R0, or have little effect, but would affect the rate of disease increase in the model. A removed state is indirectly achieved by roguing or harvesting. However, plants _can_ naturally move from the I to the R states without human activity, although the infectious period in the plant may be long enough that little of this transition occurs (for some diseases for some crops, such as for annuals). But this all deserves some attention in the Discussion. That is, what conclusions could be affected by use of SI rather than SEIR plant model? I realize that an SI type model is also used for the two vector populations, but this makes perfectly sense for NPT plant viruses (no latent period in the vector and aphids lose the plant virus very quickly and become virus-free again).

The authors put a big emphasis on roguing (removing of diseased plants). This makes sense for some crops, such as trees or high-value ornamentals. However, roguing is never (or rarely) done for many other crops, such as arable or field crops (maize, wheat, soybeans), or annuals in general. Moreover, roguing only sometimes is done in conjunction with replacement with a new disease-free plant (what the model is assuming). In many cases, diseased trees are simply removed without replacement. Based on the authors’ SI model, these latter scenarios (leading to a nonconstant total plant host population) are not considered. Of course, all models are simplifications, but the authors should address their simplifications in their Discussion (i.e., future needed work).

Reviewer #2: This paper theoretically explores the effects of interference competition between two vector species on the spread of a plant virus. One vector species is resident and the other is transient on the host plant. Only the resident species is vulnerable to insecticides. The resident species can repel the transient species either directly (competition for space) or indirectly (by the emission of volatiles compounds by the plant). Technically, the authors consider a SIR-like ODE system composed of 6 six equations. The originality of their work is to consider two vector species and their interferences through functions that depend negatively on the (relative) number of resident vectors per plant to describe direct (respectively indirect) interference mechanisms. Both visitation and emigration interference are considered. The authors show that increasing irrigation and/or fertilization, and therefore increasing the host capacity of the plant, has a unimodal effect on disease incidence or its proxy (the basic reproductive number R0) when interference is indirect. Specifically, a moderate use of irrigation and/or fertilization may maximize interference between vectors and therefor minimize disease incidence (Fig. 3A). Similarly, a minimum value of R0 is obtained for intermediate values of pesticide-induced resident mortality (Fig. 3B). Interestingly, the optimal pesticide application rate is positively related with the irrigation/fertilization rate (Fig. 4A). This means that eradication of the disease can be achieved either by high irrigation / fertilization and high use of pesticides, or by low irrigation / fertilization and low use of pesticides.

The manuscript is very clear and well written. I spotted only one typo: line 337, summarizeD. The mathematical analysis of the model is clear and rigorous.

I have only one remark: in the abstract, the authors refer to pesticides promoting the spread of NPT viruses as a counter-intuitive result, while it is a direct consequence of a model assumption. Namely, pesticides only kill resident vectors, not transient vectors. Since the authors refer to this lack of effect of pesticides on transient vectors as being well known, and since transient vectors are the main driver of disease spread, the result is rather expected. I suggest deleting “counter-intuitively” in the abstract. By the way, since it is also well know that pesticides can kill non-target species, I would like to have more explanations of why transient vectors are spared from pesticides in the Discussion.

Reviewer #3: a good modelling study

**Have the authors made all data and (if applicable) computational code underlying the findings in their manuscript fully available?**

Reviewer #1: Yes

Reviewer #2: Yes

Reviewer #3: Yes

PLOS authors have the option to publish the peer review history of their article (what does this mean?). If published, this will include your full peer review and any attached files.

Reviewer #1: No

Reviewer #2: No

Reviewer #3: No

Figure Files:

Data Requirements:

Reproducibility:

References:

---

## [Editor Report · Decision Letter 1]

7 Dec 2021

Dear Dr. Cunniffe,

We are pleased to inform you that your manuscript 'Modelling interference between vectors of non-persistently transmitted plant viruses to identify effective control strategies' has been provisionally accepted for publication in PLOS Computational Biology.

Best regards,

Konstantin B. Blyuss

Guest Editor

PLOS Computational Biology

Virginia Pitzer

Deputy Editor-in-Chief

PLOS Computational Biology

---

## [Editor Report · Acceptance letter]

21 Dec 2021

PCOMPBIOL-D-21-01416R1 

Modelling interference between vectors of non-persistently transmitted plant viruses to identify effective control strategies

Dear Dr Cunniffe,

I am pleased to inform you that your manuscript has been formally accepted for publication in PLOS Computational Biology. Your manuscript is now with our production department and you will be notified of the publication date in due course.

With kind regards,

Livia Horvath
